# A data flow process for confidential data and its application in a health research project

**Samantha S. R. Crossfield**[1]*, **Kieran Zucker**[2], **Paul Baxter**[3], **Penny Wright**[2],
**Jon Fistein**[1], **Alex F. Markham**[1,2], **Mark Birkin**[1], **Adam W. Glaser**[1,2‡], **Geoff Hall**[1,2‡]

**1** Leeds Institute for Data Analytics, University of Leeds, Leeds, United Kingdom, **2** Leeds Institute of Medical Research at St James's, University of Leeds, Leeds, United Kingdom, **3** Leeds Institute of Cardiovascular and Metabolic Medicine, University of Leeds, Leeds, United Kingdom

‡ AWG and GH are joint senior authors on this work.
* s.crossfield@leeds.ac.uk

## Abstract

### Background

The use of linked healthcare data in research has the potential to make major contributions to knowledge generation and service improvement. However, using healthcare data for secondary purposes raises legal and ethical concerns relating to confidentiality, privacy and data protection rights. Using a linkage and anonymisation approach that processes data lawfully and in line with ethical best practice to create an anonymous (non-personal) dataset can address these concerns, yet there is no set approach for defining all of the steps involved in such data flow end-to-end. We aimed to define such an approach with clear steps for dataset creation, and to describe its utilisation in a case study linking healthcare data.

### Methods

We developed a data flow protocol that generates pseudonymous datasets that can be reversibly linked, or irreversibly linked to form an anonymous research dataset. It was designed and implemented by the Comprehensive Patient Records (CPR) study in Leeds, UK.

### Results

We defined a clear approach that received ethico-legal approval for use in creating an anonymous research dataset. Our approach used individual-level linkage through a mechanism that is not computer-intensive and was rendered irreversible to both data providers and processors. We successfully applied it in the CPR study to hospital and general practice and community electronic health record data from two providers, along with patient reported outcomes, for 365,193 patients. The resultant anonymous research dataset is available via DATA-CAN, the Health Data Research Hub for Cancer in the UK.

### Conclusions

Through ethical, legal and academic review, we believe that we contribute a defined approach that represents a framework that exceeds current minimum standards for effective

**Data Availability Statement:** We provide a template version of the data flow protocol in the Supporting Information. The CPR data flow protocol and project information are available on

the LIDA website [https://lida.leeds.ac.uk/
comprehensive-patient-records-2/cpr_approvals/].
The CPR dataset can be accessed through
submitting a request to the health data research
hub DATA-CAN, who provide secure research
access, given the potentially sensitive nature of the
health data [https://web.www.healthdatagateway.
org/dataset/ce4582a8-0985-46c6-b95f-
29a5de862d4a].

**Funding:** SSRC, KZ, PB, PW, JF, AWG and GH
were supported by Macmillan Cancer Support
(106451) [https://www.macmillan.org.uk/]. SSRC
and AFM were supported by the Medical Research
Council Leeds Medical Bioinformatics Centre (MR/
L01629X) [https://gtr.ukri.org/projects?ref=MR%
2FL01629X%2F1]. Mark Birkin was supported by
the Economic and Social Research Council (ESRC)
Consumer Research Data Centre (ES/L011891/1)
[https://www.cdrc.ac.uk/]. The funders had no role
in study design, data collection and analysis,
decision to publish, or preparation of the
manuscript.

**Competing interests:** The authors have declared
that no competing interests exist.

pseudonymisation and anonymisation. This paper describes our methods and provides supporting information to facilitate the use of this approach in research.

## Introduction

There are significant barriers to the linkage and use of routinely captured healthcare data in research to generate novel insights. Data capture in discrete, 'siloed' systems renders the necessity of data linkage in research. However, linkage brings issues, such as how to handle varied data structures and maintain data security while enabling appropriate access [1]. Ethical and legal requirements may necessitate complex processes in data handling, which may be burdensome [2, 3]. In the European Union, the General Data Protection Regulation (GDPR) sets conditions for the lawful processing of personal and "special category" data that relates to an identified / identifiable individual [4]. In the UK, the Data Protection Act 2018 and common law duty of confidentiality apply [5, 6]. To disclose confidential information, collected during healthcare delivery, for secondary use in research, requires a legal gateway such as consent, public interest, a legal obligation or approval by the UK Secretary of State for Health [7]. Unprecedented rates of data sharing in response to the current pandemic are yielding important insights into COVID-19, however it is uncertain how this might change the landscape for all research [8, 9]. Further, there remains public concern, and weaknesses in public engagement, to overcome [10–12].

While secure data sharing approaches exist for the use of identifiable confidential data by approved users [13–17], secondary use in research often requires data that is, for the above ethico-legal reasons, "anonymous in such a manner that the data subject is not or no longer identifiable" beyond 'reasonable effort' [4, 7, 18, 19]. A clear and practical process is needed to link datasets and create a research dataset that is no longer considered confidential or personal. Data linkage and anonymisation procedures can be used to create an irreversibly linked, non-personal, dataset from two or more identifiable sources. Data are linked and the relationship between individuals and data are removed so that it is beyond reasonable effort to identify individuals from the data [20]. Anonymisation approaches include the removal of direct identifiers and information that is rare (either by itself or in combination with other information) [21]. Data is reduced in granularity to reduce the risk of sharing identifiable or identifying information, which can be checked using disclosure control tools such as QAMyData [22], though there may be a trade-off with utility.

Linkage and anonymisation procedures can be applied by data providers ('at-source') or "trusted third parties", although each has potential practical and security limitations. At-source linkage generally occurs where the source holds multiple datasets and the approval necessary to perform such linkage. It prevents identifying data from crossing organisational boundaries, however this also prevents authorised linkage between datasets held by multiple sources–which is often required in research. At-source research platforms have developed recently to facilitate rapid COVID-19 research, but the long-term funding and sustainability of such approaches is uncertain [23]. Alternatively, a third party may receive identifiable data to link and anonymise. In the UK, the National Data Guardian recommends that a "safe-haven" performs this role, acting as an "honest broker" [24, 25]. The NHS Digital linkage service [26] is suitable for health data that is identifiable or carries a risk of re-identification through reasonable means [25]. In response to the COVID-19 pandemic, key UK funders have worked to increase research scale and infrastructure, enabling access to National Core Data assets

through Trusted Research Environments (TREs), to enable COVID-19 research [27]. However, the transfer of personal data to a third party for linkage and anonymisation carries the risk of disclosure. It may also be legally restricted by the purposes for which the data was collected, as a legal reason (such as public interest during the COVID-19 pandemic) is required for third party processing of personal data. Further, nationally funded third parties may focus efforts on datasets commonly required for core research, which would not address all research requirements.

Pseudonymisation approaches that de-couple data from identifying information while maintaining a link between these that is accessible only by authorised parties [28, 29] offer a means to enable authorised data linkage. In pseudonymisation, directly identifying data that would otherwise be used in linkage can be replaced with a "digest" (linkable pseudonym) [30]. Recognised hashing algorithms such as SHA-2 and SHA-3 can be implemented in software such as Python or OpenPseudonymiser, to generate linkage digests from data strings, while affine cipher-based encryption may be suitable for image data [31–34]. These digests are anonymous in the context of the recipient. However, the data provider can link reversible pseudonyms back to an individual, so they remain classed as personal data [4, 29]. An approach whereby the digest is re-pseudonymised by the recipient [31], prevents backward-engineering or re-identification of the digests by any single party. However, the digests retain their status as personal data. Multiparty cryptographic protocols, such as garbled circuit approaches, can aim to prevent data linkage reversibility, but are computationally expensive [35, 36].

We aimed to describe an approach using at-source pseudonymisation with third party data linkage and anonymisation, and apply this as a case study to health data from two sources to produce an anonymous person-level research dataset. We aimed to present the practical steps involved through describing the protocol and exemplar in order to provide a framework for other projects using personal data, to either generate a pseudonymous or anonymous dataset. This may assist the usability of such data in addressing important research in industry and society.

## Materials and methods

A data flow protocol was developed and implemented as part of the *Macmillan* Cancer Support funded Comprehensive Patient Records (CPR) study, conducted at the University of Leeds in collaboration with the Leeds Teaching Hospitals NHS Trust (LTHT) and The Phoenix Partnership Leeds UK (TPP), a clinical systems company. Here we describe the method of its development, including external validation of its ethical and legal compliance, and implementation. The data flow protocol and its implementation are described in the Results.

### Protocol development

The data flow protocol was developed with input from patients, informaticians, ethicists, legal experts and colleagues from *Public Health England* and *NHS Digital*. The patient viewpoint was incorporated throughout, with two named investigators being patients who attended two-monthly meetings. The protocol and study design were presented to local research advisory groups (Cancer Research UK Leeds Centre's Public and Patient Involvement in Research Group and the Research Advisory Group of the University of Leeds Patient Reported Outcomes Group [37]), which viewed the data handling policies positively. Legal advice was sought in order to validate that the protocol met the necessary ethical and legal requirements. The University of Leeds IT department and legal team undertook a Privacy Impact Assessment and sought external independent legal counsel from the Queen's Counsel, concluding that the protocol was robust and appropriate.

## Separation of duties

A supporting separation of duties protocol was defined to establish physically separate stages of data handling (data provision; the stages of data linkage, preparation and review of derived datasets; data analysis) between teams and to prevent research access to data prior to the completion of anonymisation. This ensured that: each data provider conducted data extraction and the creation of matching pseudonymous digests; a third party provided separate teams for linkage and digest destruction, anonymisation and data preparation / minimisation, and data review and transfer; and the research team conducted analyses on the resultant anonymised research dataset. This prevented unauthorised handling of identifiable information or data with reasonable potential for re-identification.

## Data handling organisations in the case study

GP practice and community, and hospital data were sourced from TPP and LTHT for the CPR dataset. TPP provides SystmOne, a clinical system used by 7,000+ health and social care organisations in the UK to maintain ~50 million health records. TPP deliver ResearchOne [38] as a programme with national ethics approval (11/NE/0184) to facilitate research data provision. Organisations using SystmOne (e.g. GP practice and community) can opt in to providing non-identifiable data for research purposes to ResearchOne, which currently contains >8 million patient records. Organisations can also record patient opt-outs, to exclude patient data from the ResearchOne database. LTHT is one of the largest teaching hospitals in Europe and contains the Leeds Cancer Centre. The main LTHT health record system, PPM+, has previously been described and contains detailed diagnostic and management data on ~250,000 cancer patients [39]. LTHT have internally linked these clinical records to other LTHT informatics systems, including those containing financial data. LTHT also records research opt-outs, to exclude patient data from LTHT research data extracts.

In the UK, a national data opt-out service enables patients to exclude their confidential patient information from secondary use such as research. Both TPP and LTHT excluded data from patients that had opted out nationally.

Discussion with colleagues from NHS Digital determined that the Data Analytics Team (DAT) at the Leeds Institute for Data Analytics (LIDA) at the University of Leeds were an appropriate third party. The DAT provided a third-party linkage, anonymisation and data transformation service and a secure data platform for NHS Data Security and Protection Toolkit (DSPT)-compliant data handling [40]. The service was developed to provide secure information handling processes, technical infrastructure and application development for the Medical Research Council Medical Bioinformatics Centre and the Economic and Social Research Council Consumer Research Data Centre [40, 41].

## Protocol review

The protocol underwent a series of reviews in order to ensure that the data transfer could be conducted independently of a safe haven. In England and Wales, the NHS Health Research Authority (HRA) Research Ethics Committee (REC) reviews whether research is ethical and the HRA Confidentiality Advisory Group (CAG) reviews any access to confidential data without consent. The data flow protocol received ethical approval from the HRA REC (IRAS project ID 188345 REC reference 16/NE/0155) and was approved by the data providers and the third party. Informed patient consent was not sought as the data were analysed anonymously. We submitted to review by CAG explaining that we believed the confidential data processing by the data providers was according to their processing agreements, that the data was anonymous in context for the third party and research team received anonymous data. The CAG determined that approval under the

National Health Service Act Section 251 was not required given these factors, following assurance from the data providers on the at-source anonymisation procedures applied [42]. The third party sought review of the proposal by the Queen's Counsel. This confirmed the opinion that the data received by the third party and used by its researchers could be characterised as 'anonymous' and not 'personal' data. The received digests were deemed anonymous in context, and following the deletion of the salt used in their re-pseudonymisation, the final digests were deemed anonymous: although person-level, they would not contribute to identification [20].

## Data flow in the case study

Implementation of the protocol began with the data providers and ends with research output dissemination. Each data provider selected the records that were eligible for linkage from PPM+ or ResearchOne, using the project's cohort selection criteria (S1 Table), through their own computerised processes in accordance with the LTHT Fair Processing Notice and ResearchOne Database Protocol, respectively [43, 44]. The research team provided the data providers with standard operating procedures to ensure a standardised approach. The Caldicott Guardian of one provider produced an encrypted project-specific "salt" file via the OpenPseudonymiser website [31] and shared this with the analyst team in both organisations. The salt was code created using OpenPseudonymiser from a project-specific alphanumeric string. Digests only matched where they are created using the same identifiers (in the same format) and salt, which prevents unauthorised linkage. The analysts combined the project salt with the agreed patient identifiable inputs and used OpenPseudonymiser to produce linkable pseudonymous digests using the secure hash algorithm (SHA-256) [31]. These were encrypted and provided to the DAT using a secure file transfer protocol (SFTP). A designated analyst produced a list of matches and returned these to the providers before locally destroying the received and matched files. The providers produced data for this linked cohort, using the project's data items list. The data providers additionally applied their standard anonymisation processes prior to data transfer, such as replacing postcodes with sector-level postcode or the corresponding Index of Multiple Deprivation score [45].

The extracts were encrypted and transferred via SFTP to the DAT, containing digests that were anonymous in the context of the DAT. A DAT analyst re-pseudonymised the digests using a DAT project-specific salt which was then destroyed (rendering the digests anonymous in all contexts). A DAT analyst then linked the extracts and performed other transformations to produce the anonymous research dataset. The transformations aimed to address the risk of re-identification inherent in data linkage by undertaking data minimisation steps, such as replacing the patient's month and year of birth with an age-band and converting dated diagnostic data into a binary indicator of prevalence within specific time-frames. This helped to avoid the risk of the linked dataset containing "digital fingerprints" [46]. The processing applied to some common data types are listed in Table 1. Controlled research data access was provided to researchers over a University network under a user agreement detailing their data processing obligations. This included sanctioning against any attempts at re-identification. Access in a project-specific, firewalled environment in compliance with the NHS DSPT [47] was contingent upon providing evidence of completing annual training in advanced information security. An agreement was defined to ensure that the DAT handled all data ingress and egress, providing an independent check against the ethical and legal requirements.

## Results

### Data flow protocol

Informed by patients, informaticians, ethicists, legal experts, clinicians and colleagues from Public Health England and NHS Digital, we developed a data flow protocol resulting in an

**Table 1. Common data types and the data processing steps applied in the Comprehensive Patient Records project.**

| Data Type | Data Processing Steps |
|---|---|
| Patient Name | Excluded at source |
| NHS Number | An input variable in pseudonymous digest creation (performed by the data providers) |
| Date of Birth | Transformed into age at first cancer diagnosis / matched index date, in age-bands (<1 years, 1–4 years and 5-year bands thereafter until 80–100) |
| Date of Death | A Boolean indicator of survival status as known to the data source was provided |
| Postcode | Reduced to postcode sector only e.g. LS1 5, or mapped to Index of Multiple Deprivation score [45] |
| Diagnostic Codes | Aggregated to a binary yes/no for prevalence of disease or disease groups in time periods pre- and post- cancer diagnosis or matched index date |
| Prescribing Data | Mapped to annual cost per patient; aggregated to a Boolean for presence or absence of diabetes drug classes |
| Sex | No processing applied |
| Ethnicity | No processing applied (coded using national code-lists) |
| Date last seen by primary care team | No processing applied |

irreversibly linked anonymous dataset (S1 File). The protocol utilised OpenPseudonymiser, as an open source application for converting an input field into digests using the Secure Hash Algorithm 2 [31, 48]. The data providers agreed on the direct identifiers and a salt to be used to create linkable digests. The inputs were formatted using an agreed approach. Following the protocol, the data providers transferred these digests to a third party who advised on the matching digests, for which records the data providers each produced a dataset with pseudonymous digests. Upon receiving these datasets, the third party re-pseudonymised the digests using a unique salt that was then destroyed to render the digests anonymous, and linked the datasets using the new digests. The third party performed further transformations to ensure the anonymisation of the merged dataset. All data transfers were in encrypted format. These steps are detailed in Table 2 and a template version of the data flow protocol, which defines the data flow and technical procedure, and provides a supporting glossary and worked examples of the pseudonymisation approach (S1 File).

## Case study

The data flow protocol was utilised by the CPR study to link anonymised routinely collected health data from hospital, GP practice and community electronic health records in England (Fig 1). Fig 1 depicts the parties involved in the case study, they key actions that they performed, and the transfer of, and access to, data. A separation of duties procedure allocated the data handling functions in the data flow protocol to specific parties to ensure separation of the pseudonymisation and anonymisation steps and to define the digests handled by each party. Two organisations undertook the data provision: LTHT and TPP, as described in the Methods. The data providers shared a project-specific salt file and followed a standard procedure to produce patient digests. LTHT identified 102,763 patients with a cancer diagnosis, defined either via clinical review ("gold standard") or through the recorded cancer status, ICD-10 code, and definitive disease phase, and 287,564 matched non-cancer patients (Fig 2). TPP provided digests for the ResearchOne database (approximately 8 million). The DAT in LIDA, independent of the research team, identified 140,462 matching digests. The data providers returned anonymised data for this cohort, which were linked within LIDA. The digests were then re-pseudonymised and the salt used was destroyed in order to irreversibly break the link back to

**Table 2. Summary of actions defined in the data flow protocol and the parties involved in each step.**

| Step | Action Description | Organisation |
|---|---|---|
| 1 | Create and share a hashed project-specific salt (SALT1) | Data providers |
| 2 | Determine records eligible for linkage | Data providers |
| 3 | Use agreed fields and SALT1 to generate project-specific digests (PSD1s) for these records | Data providers |
| 4 | Transfer PSD1s to the linkage party | Data providers |
| 5 | Compile a list of matching PSD1s, return to data providers | Third party |
| 6 | Delete any locally-held PSD1s | Third party |
| 7 | Creation of at-source anonymised datasets; transfer to the linkage party | Data providers |
| 8 | Creation of a project-specific salt (SALT2) and replacement of PSD1s with a second digest (PSD2) | Third party |
| 9 | Linkage of datasets to produce the research dataset (RD) | Third party |
| 10 | Authorised research access to the RD is granted, in a trusted research environment | Third party |
| 11 | Analysis of the RD; research output generation | Research team |
| 12 | Outputs screened for risk of re-identification and reviewed against ethical and governance requirements prior to authorised release | Third party |

the data provider datasets. Further aggregation and transformations were also applied to minimise the risk of re-identification following data linkage and to produce the research dataset in the format desired by this research project.

## Discussion

We believe that this data flow process meets the urgent need for a lawful, ethical and practical framework for personal data handling in research. It incorporates best practice mechanisms and addresses ethical and legal requirements to enable the flow of person-level data for

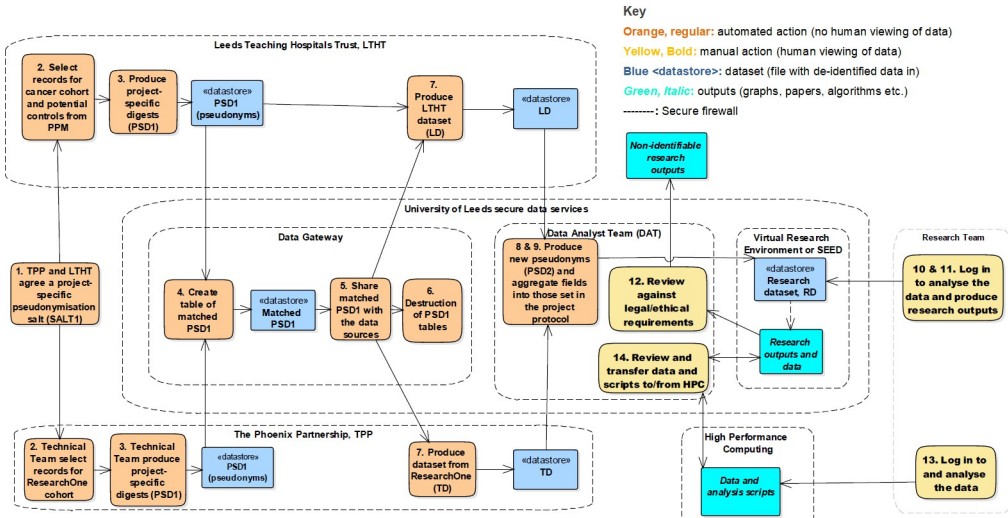

**Fig 1. Diagram of the parties involved in the case study and the actions performed during the process of data flow, linkage and access, as defined using the protocol for linkage and anonymisation of data.**

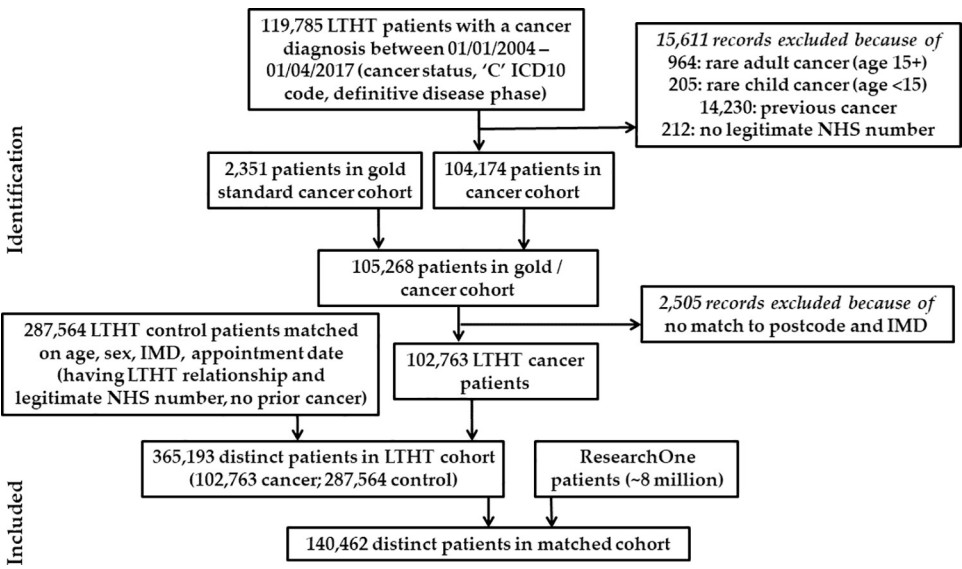

**Fig 2. Flow chart of the patient cohort selection process for the case study, defining the number of patient records in each stage of selection and linkage.**

research. This data flow process offers one approach to support data dissemination at scale for secondary uses, by producing an at-source anonymised dataset that is irreversibly linked using accepted cryptographic standards. It can also be easily adapted for use in projects that have approval to utilise reversible pseudonymisation. The protocol is made available to inform other projects, so that it may facilitate access to datasets and data-based research. Further its application in the case study has made available an anonymous research dataset that may be invaluable in future research.

Ethical data handling is particularly crucial for personal healthcare data and so we have exemplified the process with a case-study linking and anonymising such data. The successful implementation by this CPR study clarifies the efficacy of the process and how to practically apply it to real-world issues. The CPR dataset has been incorporated as an exemplar research dataset by DATA-CAN, the Health Data Research Hub for Cancer in the UK [49]. The data flow protocol has already been adopted locally by projects including the Yorkshire Specialist Register of Cancer in Children and Young People [50]. The approach may be adaptable for international applicability. Such linkage projects may facilitate in unlocking patient benefit from the wealth of information currently available within routinely collected data.

## Process review

Internal and external review of the process demonstrated that the protocol ensures appropriate handling of all ethical considerations and stakeholder perspectives. It met or exceeded the current UK legal requirements and advice on data handling, including those defined in The Information Governance Review and the NHS DSPT standards [25, 47]. The perspectives of the data subject were embedded through patient and public involvement and engagement in all stages of the process, including protocol development, review and implementation. We engaged data providers to ensure that the protocol provides the necessary practical guidance to produce linkable datasets. This brought transparency to the process, which in turn enabled reproducible reporting of the study population and informs the interpretation of research [51]. A privacy impact assessment, now known as a data protection impact assessment (DPIA), was

performed by the University of Leeds IT Assurance and legal team, given the involvement (at-source) of personal data. This independently assessed and informed the process before deeming it appropriate [52]. Future projects adopting a similar protocol may anticipate being assessed as 'low risk' during their DPIA given the view expressed in the CPR DPIA and by the Queen's Counsel, that the data was not personal in the context of the third party and researchers. We undertook significant review so that the process might be adapted to future projects as one way to adhere with the current standards and legal requirements, and address the ethical and privacy concerns that arise in data use and linkage generally.

## Cross-accreditation of data linkage and anonymisation processes

In the UK, the Information Governance Review [25] describes "accredited data safe havens" as the appropriate locations for the linkage and anonymisation of personal confidential data or that which could potentially identify individuals and is linked for limited disclosure/access. TREs and health data hubs host researcher access to core datasets, however, research projects such as the CPR often require a bespoke data flow. Unlike the clear researcher accreditation requirements that are listed for TREs [27], the criteria for an organisation to become accredited as a safe haven for such data processing are unclear. This is problematic as this is required for an organisation to be appropriate to undertake linkage or anonymisation processes and to meet bespoke research needs. A formal list of requirements for accreditation for linkage and / or anonymisation is required. Further, the former should extend to all data linkage scenarios to ensure comprehensive compliance with ethical requirements.

We propose an approach of cross-accreditation between organisations that adopt our data flow process or conduct similar data handling. We propose a system whereby institutions independently evaluate each other's processes and infrastructure, nationally and internationally. This would foster expert practice and enable institutions in academia and industry to become 'cross-accredited data safe havens'. Through increased transparency, this would reduce the uncertainty and variation in current practice. It could be initiated using the open and reproducible criteria that we propose in Table 3, and would evolve through practice. Datasets developed securely by such safe havens could subsequently feed into TREs or health data research hubs to facilitate wider access by accredited researchers.

**Table 3. Recommended minimum criteria for incorporation into a checklist for data safe haven cross-accreditation.**

| Recommended Minimum Criteria |
|---|
| Protocols and work instructions that incorporate any relevant ethical and legal measures |
| Demonstrate compliance with the NHS DSPT standards when handling health data (UK health data- specific) |
| Robust data access control |
| User training in advanced information security |
| An information security management system with an assigned data protection officer and information governance management group oversight |
| Procedures for data classification, measuring identifiability, anonymisation, risk assessment and privacy impact assessment |
| Demonstrable adherence to procedures for appropriate auditing of controls |
| Where data being linked are confidential or could potentially identify individuals and are linked for purposes of limited disclosure/access: demonstrate reasonable data stewardship (which in the UK is defined in The 2013 Information Governance Review [25]) and provide an environment, controls or sanctioned agreements for limited disclosure/access |
| Where provision is also provided for analysis using identifiable data: remote access and controlled data ingress and egress; third-party review of outputs using data non-disclosure principles prior to dissemination; availability of 'safe room' with restricted access as appropriate |

## Strengths and limitations

The protocol utilises the concepts of at-source pseudonymisation and third party anonymisation of the linkage digests to enable non-reversible data linkage. This approach brings advantages without compromising either matching accuracy or computer processing time, as would be the case with cryptographic approaches such as garbled circuits to complete linkage at scale [35]. It reduces the risk of information disclosure by external attack, as the linked research dataset does not link back to digests held by the data providers. Two salts are used to produce the digests and no party holds both, as would be required for digest re-identification. Even in the case of malicious attack, it is not reasonable to anticipate digest re-identification in comparison with some honest broker approaches [53]. Anonymisation using a third party and a separation of duties procedure prevents data or digest identification through collusion between the data providers and research team, providing protection above pseudonymisation approaches. The digest anonymisation approach used strengthens the linkage process, although it must be coupled with processes that mitigate the risk of identification through the rest of the dataset (i.e. data that is shared but not used in the linkage process). Such processes should be selected to ensure that data are anonymous beyond reasonable effort, while maintaining wherever possible the dataset utility in relation to addressing the research question [19]. A party, e.g. the data provider or third party, must apply these processes under the appropriate consent and legal frameworks. The protocol acknowledges that data linkage may necessitate further assessment of risk and post-linkage anonymisation or aggregation steps, to produce a research dataset that is richer than unlinked data while being practically protected from internal disclosure of information. As part of a governance framework, including data access limitations (including prevention of concurrent access to auxiliary information), processing agreements and pre-dissemination disclosure review, the approach in the data flow protocol can assist in preventing unauthorised attempts at re-identification [30, 54, 55].

Our case study demonstrated successful implementation of the protocol. The case study employed a third party that acted independent of the research team but was within the same organisation (University of Leeds), which minimised the data disclosure risks inherent in otherwise involving further organisations in data handling. The third party contributed to secure data flow by verifying the involved parties and reviewing the approvals in place for data handling [17]. Alternative approaches to party verification could be considered, alongside or instead of this being performed by a third party [56]. In the case study, the cancer status of patients as defined using hospital data was not shared with the GP and community data provider owing to the inclusion of non-cancer patients. As almost all patients in the UK are registered with a GP, we believe that linkage did not provide the hospital data provider with additional information about the data subjects either. Studies of especially sensitive or contentious topics may similarly consider including a non-disease / event cohort to prevent data providers from learning a sensitive status about the data subjects. However, for studies that would not analyse data on such a cohort, this brings additional data flow, which should be minimised by immediate third party deletion of any data received in the data extracts for this surplus cohort.

The involvement of specific data, data providers or recipients may necessitate adaptations to the protocol. For example, multiple providers may not share matching fields for linkage, in which case one or more providers may act as a 'bridge' by providing multiple digests, each to be matched with those from a different provider. Other schemes could be considered for reaching agreement on the salt used by data providers in pseudonymisation [56]. Controls may need to be applied to the environment in which the dataset is held, for example where data access agreements do not mitigate against linkage with auxiliary information, through

which 'jigsaw re-identification' may occur [57]. This may be at a loss of flexibility in the environment or researcher autonomy. The transferability of our data flow approach may be limited where there are additional legal requirements, industrial standards and organisational agreements applicable to different data types (e.g. human tissue), institutes and countries. Amendments may be necessary in transferring the described approach to other contexts or jurisdictions, which may require additional ethical and legal scrutiny and stakeholder involvement. However, the protocol template and steps described should aid in considering the relevant issues and developing a tailored protocol. The protocol can even be adapted if other contexts necessitate using other approaches to pseudonymisation or anonymisation [58, 59]. For example, if a project had approval to use pseusonymised data in order to relay study results to the participants, the protocol could be adapted by retaining the third party salt. Using the template, with adaptations, will efficiently aid in appropriately considering the wider information security and governance framework, including secure data transfer and access restrictions.

In the case study, the third party did not perform a quantitative assessment of the residual risk of re-identification in the final dataset. However, the final list of variables was constrained and was reviewed including by the HRA, CAG, Queen's Counsel and University of Leeds legal team and IT Assurance. Projects using different datasets will require consideration of re-identification risk following data linkage. However, we list the data processing applied in the case study to common data types (Table 1), which should aid in determining any appropriate steps. Tools such as sdcMicro could aid the third party in checking for any disclosure risk brought through the linking of variables [60]. Such work may bring resource cost and delay research access. Researchers should be informed of all data processing steps in order to be able to consider any impact on the statistical properties of the dataset that may bias analyses [51].

In our approach, anonymous digests were created by third party re-pseudonymisation of the digests, with the salt used being promptly destroyed. The third party could have alternatively replaced each digest with a random ID, but given the difficulties in true randomisation this random ID would still then require pseudonymising with a salt that is destroyed, so such an approach would have introduced extra steps into the process. While in this study the digests were deemed anonymous in context of the third party (e.g. by the Queen's Counsel) prior to re-pseudonymisation and salt destruction, it is hoped that in time there will be further clarification, in national and international contexts, of the legal stance on such data. This is particularly necessary given the unprecedented rise in efforts to facilitate data access in response to COVID-19 [23, 61].

## Conclusions

We contribute a data flow protocol for linkage and anonymisation and a description of its successful utilisation in a case study to produce a dataset available for use in research. The presented risk-mitigating approach to data flow offers a practical solution to commonly encountered issues of data handling. It employs factors to remove the link to named individuals from person-level data. It incorporates best practice and provides adaptable steps for handling data in accordance with the current ethical and legal framework in the UK and European Union, but has potential global application. This flexible and adaptable protocol may facilitate the move toward widespread utilisation of the growing volumes of data available. We also proposed a cross-accreditation approach and recommended criteria that may support organisations in adopting appropriate data handling practices and offering public reassurance.

## Supporting information

**S1 Table. Inclusion criteria for patient selection for the Comprehensive Patient Records project.**
(DOCX)

**S1 File. Data flow protocol template.**
(DOCX)

## Acknowledgments

This work uses data provided by patients and collected by the NHS as part of their care and support. We acknowledge the guidance and input from the patient perspective from the CPR lay co-investigators, Barbara Woroncow and David Wilkinson. We acknowledge the views received from the Cancer Research UK Leeds Centre's Public and Patient Involvement in Research Group and the Research Advisory Group of the University of Leeds Patient Reported Outcomes Group. We acknowledge the data handling expertise of Adam Keeley from the Data Analytics Team at the Leeds Institute for Data Analytics. We acknowledge legal counsel from Adrian Slater (University of Leeds Legal Advisor) and Robin Hopkins QC.

## Author Contributions

**Conceptualization:** Adam W. Glaser, Geoff Hall.

**Data curation:** Samantha S. R. Crossfield, Kieran Zucker.

**Funding acquisition:** Adam W. Glaser, Geoff Hall.

**Investigation:** Samantha S. R. Crossfield, Kieran Zucker, Paul Baxter, Penny Wright, Adam W. Glaser, Geoff Hall.

**Methodology:** Samantha S. R. Crossfield, Kieran Zucker, Paul Baxter, Penny Wright, Jon Fistein, Adam W. Glaser, Geoff Hall.

**Writing – original draft:** Samantha S. R. Crossfield, Kieran Zucker.

**Writing – review & editing:** Samantha S. R. Crossfield, Kieran Zucker, Paul Baxter, Penny Wright, Jon Fistein, Alex F. Markham, Mark Birkin, Adam W. Glaser, Geoff Hall.

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
