## [Decision Letter · Decision Letter 0]

8 Oct 2021

PONE-D-21-29276A data flow process for confidential data and its application in a health research projectPLOS ONE

Dear Dr. Crossfield,

Thank you for submitting your manuscript to PLOS ONE. After careful consideration, we feel that it has merit but does not fully meet PLOS ONE’s publication criteria as it currently stands. Therefore, we invite you to submit a revised version of the manuscript that addresses the points raised during the review process.

We look forward to receiving your revised manuscript.

Kind regards,

Pandi Vijayakumar, Ph.D

Academic Editor

PLOS ONE

Journal Requirements:

"We would like to acknowledge the funding and support from Macmillan Cancer Support (106451), the Economic and Social Research Council (ES/L011891/1) and Medical Research Council (MR/L01629X)."

"SSRC, KZ, PB, PW, JF, AWG and GH were supported by Macmillan Cancer Support (106451) [https://www.macmillan.org.uk/]. SSRC and AFM were supported by the Medical Research Council Leeds Medical Bioinformatics Centre (MR/L01629X) [https://gtr.ukri.org/projects?ref=MR%2FL01629X%2F1]. Mark Birkin was supported by the Economic and Social Research Council (ESRC) Consumer Research Data Centre (ES/L011891/1) [https://www.cdrc.ac.uk/]. The funders had no role in study design, data collection and analysis, decision to publish, or preparation of the manuscript."

Additional Editor Comments:

Based on the comments of the reviewers, I recommend major revision.

Reviewers' comments:

Reviewer's Responses to Questions

**Comments to the Author**

1. Is the manuscript technically sound, and do the data support the conclusions?

Reviewer #1: Yes

Reviewer #2: Yes

2. Has the statistical analysis been performed appropriately and rigorously? 

Reviewer #1: Yes

Reviewer #2: Yes

3. Have the authors made all data underlying the findings in their manuscript fully available?

Reviewer #1: Yes

Reviewer #2: Yes

4. Is the manuscript presented in an intelligible fashion and written in standard English?

Reviewer #1: Yes

Reviewer #2: Yes

5. Review Comments to the Author

Reviewer #1: Linking the research datasets of patients without divulging the sensitive medical information of patients and hospitals is very difficult. This research work, in this line, has proposed a dataflow protocol for anonymous linked datasets for research purposes. It is accomplished by Comprehensive Patient Records study in Leeds, United Kingdom with data from two providers which contains the details of 3, 65, 193 patients.

The protocol design has been verified by research advisory groups.

The authors have received the legal advice for this protocol that it maintains ethical as well as legal requirements.

This protocol will be very useful during the Covid-19 situations to analyze and identify any useful patterns.

The use of OpenPseudonymiser for creating secure digests and linking the digests while preventing backward engineering seems to be very useful.

Reviewer #2: Authors presented a technological impact of trust in Social Government during COVID-19 crisis. Though, paper covers important topic, technical contribution of this paper is limited. Also, following comments need to be addressed in the revised manuscript:

-Highlight your contribution clearly in Abstract section.

-Mention the quantitative findings of this research in the abstract section.

-Fig. 1 is not explained properly.

-Also, in some figures, captions are not proper.

-Introduction section needs to be re-written to improve its quality and readability.

-Put some more light on concept of the paper and its usability to the industry and society.

-Paper needs to polish and provide a detailed explication of theoretical aspects such as conditions and theorems, and practical issues like algorithms, rules and possible applications.

-Following are some of relevant and recent references which can be referred in the revised manuscript:

Provably Secure Data Sharing Approach for Personal Health Records in Cloud Storage Using Session Password, Data Access Key, and Circular Interpolation,

Capability based outsourced data access control with assured file deletion and efficient revocation with trust factor in cloud computing,

IoT-based big data secure management in the fog over a 6G wireless network,

Efficient escrow-free CP-ABE with constant size ciphertext and secret key for big data storage in cloud,

A novel CNN based security guaranteed image watermarking generation scenario for smart city applications.

-In the conclusion section, I suggest, to highlighting the new discovery, new inventions and new aspects of the research.

-Improve overall flow of this paper for better understanding.

6. PLOS authors have the option to publish the peer review history of their article (what does this mean?). If published, this will include your full peer review and any attached files.

Reviewer #1: No

Reviewer #2: No

---

## [Author Response · Author response to Decision Letter 0]

11 Nov 2021

Thank you for the feedback and opportunity to address the comments raised. We have added our response to all points raised by the reviewers in line below, and in the attached 'Response to Reviewers' document. Line references correspond to the attached ‘Revised Manuscript with Track Changes’.

Reviewer #1: Linking the research datasets of patients without divulging the sensitive medical information of patients and hospitals is very difficult. This research work, in this line, has proposed a dataflow protocol for anonymous linked datasets for research purposes. It is accomplished by Comprehensive Patient Records study in Leeds, United Kingdom with data from two providers which contains the details of 365, 193 patients.

The protocol design has been verified by research advisory groups.

The authors have received the legal advice for this protocol that it maintains ethical as well as legal requirements.

This protocol will be very useful during the Covid-19 situations to analyze and identify any useful patterns.

The use of OpenPseudonymiser for creating secure digests and linking the digests while preventing backward engineering seems to be very useful.

Many thanks for your review and feedback.

Reviewer #2: Authors presented a technological impact of trust in Social Government during COVID-19 crisis. Though, paper covers important topic, technical contribution of this paper is limited. Also, following comments need to be addressed in the revised manuscript:

-Highlight your contribution clearly in Abstract section.

We believe that this manuscript contributes a linkage and anonymisation approach with clear steps for creating a research dataset in line with ethico-legal requirements. We also produced a research dataset that has important value for cancer research and describe how it has been made accessible.

We have now clarified this by highlighting the issue, study aim and contribution in the abstract (lines 28-30, 34-36 and 52-53).

-Mention the quantitative findings of this research in the abstract section.

The main findings of the research were that a) a data flow protocol was defined for linking and anonymising data with ethico-legal approval; and b) the data flow protocol was successfully applied in a case study in which data regarding 365,193 patients was linked and anonymised from two healthcare providers. The invaluable cancer dataset that was created is now available for use in research

We have now highlighted this more clearly in the abstract (lines 43-49)

-Fig. 1 is not explained properly.

Thank you for the opportunity to provide further clarity. The data flow protocol that is defined in the manuscript was utilised in a case study. Figure 1 depicts the parties involved in data handling in this case study: the data transfers, the linkage and anonymisation actions performed, and the research access granted to the final (linked and anonymised) research dataset.

We have now added a description of the information in Figure 1 (lines 289-290) and further clarified the caption for Fig 1 (lines 308-309).

-Also, in some figures, captions are not proper.

We have now amended the captions for both Fig 1 and Fig 2, by adding further descriptive detail (lines 307-312).

-Introduction section needs to be re-written to improve its quality and readability.

The Introduction section has been refined throughout, to clarify the issues that the manuscript addresses and to improve readability (lines 65-67, 76-80, 86-87, 93-99, 106-114, 119-123, 132-136).

-Put some more light on concept of the paper and its usability to the industry and society.

As described above, we have now defined the aim and contribution more clearly in the Abstract, and also in the Conclusions section as described in response to a further comment below. We have also elaborated on the concept of the paper and its contribution to industry and society in the Introduction (lines 132-136) and Discussion (lines 316-322).

-Paper needs to polish and provide a detailed explication of theoretical aspects such as conditions and theorems, and practical issues like algorithms, rules and possible applications.

We have now refined the flow of this paper through general changes throughout. We have also now explicitly referenced the technical detail of the data flow and worked examples of pseudonymisation that are provided in the Supporting Information (lines 279-281). The glossary of terms in the Supporting Information, which provides further explication, is also now referenced. Further, we have now highlighted the possible application of the described approach in addressing practical issues in enabling the secondary use of data for research (e.g. Abstract, Conclusions, and lines 316-322). We have made explicit that the protocol is shared and access to the CPR dataset is made available, both in order to assist in future research projects (lines 268, 321-324).

-Following are some of relevant and recent references which can be referred in the revised manuscript:

Provably Secure Data Sharing Approach for Personal Health Records in Cloud Storage Using Session Password, Data Access Key, and Circular Interpolation,

Capability based outsourced data access control with assured file deletion and efficient revocation with trust factor in cloud computing,

IoT-based big data secure management in the fog over a 6G wireless network,

Efficient escrow-free CP-ABE with constant size ciphertext and secret key for big data storage in cloud,

A novel CNN based security guaranteed image watermarking generation scenario for smart city applications.

Thank you for providing references to these manuscripts, which have now been cited as relevant (lines 74-75).

-In the conclusion section, I suggest, to highlighting the new discovery, new inventions and new aspects of the research.

The manuscript contributes a) a data flow protocol for linkage and anonymisation, b) description of a case study that successfully utilised the data flow protocol, in the healthcare domain, and information on how to request access to the resultant dataset; and c) a proposed approach and recommended criteria for cross-accreditation between organisations that handle data.

The Conclusions section has now been amended to highlight these three contributions (lines 472-481. 

-Improve overall flow of this paper for better understanding.

We have now improved the flow of this paper to aid understanding, through general changes applied throughout the manuscript.

---

## [Decision Letter · Decision Letter 1]

24 Nov 2021

PONE-D-21-29276R1A data flow process for confidential data and its application in a health research projectPLOS ONE

Dear Dr. Crossfield,

Thank you for submitting your manuscript to PLOS ONE. After careful consideration, we feel that it has merit but does not fully meet PLOS ONE’s publication criteria as it currently stands. Therefore, we invite you to submit a revised version of the manuscript that addresses the points raised during the review process.

We look forward to receiving your revised manuscript.

Kind regards,

Pandi Vijayakumar, Ph.D

Academic Editor

PLOS ONE

Journal Requirements:

Additional Editor Comments :

Based on the reviewers comments, I recommend the paper for minor revision.

Reviewers' comments:

Reviewer's Responses to Questions

**Comments to the Author**

1. If the authors have adequately addressed your comments raised in a previous round of review and you feel that this manuscript is now acceptable for publication, you may indicate that here to bypass the “Comments to the Author” section, enter your conflict of interest statement in the “Confidential to Editor” section, and submit your "Accept" recommendation.

Reviewer #1: (No Response)

Reviewer #2: (No Response)

2. Is the manuscript technically sound, and do the data support the conclusions?

Reviewer #1: Yes

Reviewer #2: (No Response)

3. Has the statistical analysis been performed appropriately and rigorously? 

Reviewer #1: Yes

Reviewer #2: (No Response)

4. Have the authors made all data underlying the findings in their manuscript fully available?

Reviewer #1: Yes

Reviewer #2: (No Response)

5. Is the manuscript presented in an intelligible fashion and written in standard English?

Reviewer #1: Yes

Reviewer #2: (No Response)

6. Review Comments to the Author

Reviewer #1: The proposed work is about the utilization of valuable health data of patients. The idea of patient confidentiality is clearly maintained by anonymizing the data and linking them together for research and other application purposes. The authors claim that, the work has potential applications not only in England and European Union as the handled datasets is pertaining to these geographical locations, but to the global scale.

But, I suggest the authors clarify in what way is this research work related to security. Apart from this, many worthwhile contributions such as

1. "An efficient anonymous authentication and confidentiality preservation schemes for secure communications in wireless body area networks"

2. "A new SmartSMS protocol for secure SMS communication in m-health environment"

3. "An efficient anonymous authentication and key agreement scheme with privacy-preserving for smart cities"

to be added to the review section of this article to make it oriented towards confidentiality and security concerns.

As a whole, this research work is a novel proposal by the authors and if the suggestions are incorporated, the paper can be considered for possible publication in your esteemed journal.

Reviewer #2: A data flow process for confidential data and its application in a health research project

is presented in this paper. Paper is revised well. It can be accepted now.

7. PLOS authors have the option to publish the peer review history of their article (what does this mean?). If published, this will include your full peer review and any attached files.

Reviewer #1: No

Reviewer #2: No

---

## [Author Response · Author response to Decision Letter 1]

9 Dec 2021

Thank you for the feedback and opportunity to address the comments raised. We have added our response to all points raised by the reviewers in line below, and as described in the attached 'Response to Reviewers'. Line references correspond to the attached ‘Revised Manuscript with Track Changes’.

Reviewer #1: The proposed work is about the utilization of valuable health data of patients. The idea of patient confidentiality is clearly maintained by anonymizing the data and linking them together for research and other application purposes. The authors claim that, the work has potential applications not only in England and European Union as the handled datasets is pertaining to these geographical locations, but to the global scale.

But, I suggest the authors clarify in what way is this research work related to security. 

As described in lines 432-435, use of the protocol encourages consideration of the wider information security and governance framework, including the adoption of mechanisms for secure data transfer and restricting data access as appropriate. We acknowledge that the research has not directly developed any security feature, but rather the research work relates to security through guiding protocol users through the steps and security measures to be considered during linkage and anonymisation exercises. 

We have therefore now clarified that the protocol relates to the linkage and anonymisation of data, rather than security directly (lines 299, 305).

Apart from this, many worthwhile contributions such as

1. "An efficient anonymous authentication and confidentiality preservation schemes for secure communications in wireless body area networks"

2. "A new SmartSMS protocol for secure SMS communication in m-health environment"

3. "An efficient anonymous authentication and key agreement scheme with privacy-preserving for smart cities"

to be added to the review section of this article to make it oriented towards confidentiality and security concerns.

Thank you for these references. We have now discussed and cited these as appropriate in the manuscript (lines 73, 112-113, 406-408, 423-424).

As a whole, this research work is a novel proposal by the authors and if the suggestions are incorporated, the paper can be considered for possible publication in your esteemed journal.

Many thanks for your review and guidance.

Reviewer #2: A data flow process for confidential data and its application in a health research project

is presented in this paper. Paper is revised well. It can be accepted now.

Many thanks for your review and feedback.

---

## [Decision Letter · Decision Letter 2]

31 Dec 2021

A data flow process for confidential data and its application in a health research project

PONE-D-21-29276R2

Dear Dr. Crossfield,

We’re pleased to inform you that your manuscript has been judged scientifically suitable for publication and will be formally accepted for publication once it meets all outstanding technical requirements.

Kind regards,

Pandi Vijayakumar, Ph.D

Academic Editor

PLOS ONE

Additional Editor Comments (optional):

Based on the comments of the reviewers, I strongly accept this paper for publication.

Reviewers' comments:

Reviewer's Responses to Questions

**Comments to the Author**

1. If the authors have adequately addressed your comments raised in a previous round of review and you feel that this manuscript is now acceptable for publication, you may indicate that here to bypass the “Comments to the Author” section, enter your conflict of interest statement in the “Confidential to Editor” section, and submit your "Accept" recommendation.

Reviewer #1: All comments have been addressed

Reviewer #2: All comments have been addressed

2. Is the manuscript technically sound, and do the data support the conclusions?

Reviewer #1: Yes

Reviewer #2: Yes

3. Has the statistical analysis been performed appropriately and rigorously? 

Reviewer #1: Yes

Reviewer #2: Yes

4. Have the authors made all data underlying the findings in their manuscript fully available?

Reviewer #1: Yes

Reviewer #2: Yes

5. Is the manuscript presented in an intelligible fashion and written in standard English?

Reviewer #1: Yes

Reviewer #2: (No Response)

6. Review Comments to the Author

Reviewer #1: (No Response)

Reviewer #2: A data flow process for confidential data and its application in a health research project is presented in this paper. It can be accepted now.

7. PLOS authors have the option to publish the peer review history of their article (what does this mean?). If published, this will include your full peer review and any attached files.

Reviewer #1: No

Reviewer #2: No

---

## [Editor Report · Acceptance letter]

10 Jan 2022

PONE-D-21-29276R2 

A data flow process for confidential data and its application in a health research project 

Dear Dr. Crossfield:

I'm pleased to inform you that your manuscript has been deemed suitable for publication in PLOS ONE. Congratulations! Your manuscript is now with our production department. 

Kind regards, 

on behalf of

Dr. Pandi Vijayakumar 

Academic Editor

PLOS ONE